



# Full-tensor gravity gradient eigenvector analysis for locating complex geological source positions

Boxin Zuo[1], Kass Mason Andrew[2], Xinagyun Hu[3], Meixia Geng[3]

[1]Hubei Key Laboratory of Intelligent Geo-Information Processing, China University of Geosciences, Wuhan 430074, China,
[2]U. S. Geological Survey, Colorado, United States
[3]Institute of Geophysics and Geomatics, China University of Geosciences, Wuhan 430074, China

* *Corresponding to:* Xiangyun Hu (xyhu@163.com)

**Abstract.** We develop an eigenvector analysis method to locate the centroids and horizontal boundaries of geological structures of full tensor gravity gradient (GGT) data. Although the boundary detection method for Bouguer gravity has been widely discussed and applied, the source location method for GGT data remains an area of active research. In this paper, we first discuss the theoretical basis and physical meaning of the eigenvector analysis on GGT data, and then a new source location method is derived. Unlike traditional potential field boundary detection, the proposed method uses eigenvector analysis to extract the source centroid information. The interference of multiple and overlapping sources and the parameter identification related with the multiple scales of the GGT eigenvector analysis are presented in the theoretical and experimental sections. Finally, the proposed method is applied to synthetic and field data.

## 1 Introduction

A gravity survey can reflect the response of density contrasts in the subsurface, such as high-density mineral deposit, or low-density oil deposit, with respect to the host or country rock. The gravity gradient anomaly as defined here is an anomalous response with respect to the background at the particular scale and magnitude of interest. For example, in geotechnical applications, a mass deficit due to a mine would result in a short-wavelength negative anomaly with respect to background geological signal of larger wavelength. Conversely, a mass excess due to an ore body would produce a positive anomaly imprinted on the background tectonic/geoid-based signals. Location of source positions is critical in the interpretation of gravity gradient data, which has been widely used in mineral exploration and resource surveys (e.g. Dransfield 1994; Mikhailov et al., 2007). Various edge detection methods for Bouguer gravity anomaly data, such as the derivative filter, analytic signal and others, have been proposed to delineate the outline of a geologic target and provide vital information for



data interpretation (Gordell 1989; Wijns et. al., 2005; Li 2006; Cooper and Cowan 2006; Foks 2013; Ma and Li 2012; Phillips 2015;).

In recent years, gravity gradient tensor (GGT) measurement devices and methods have been widely researched (e.g. Pedersen and Rasmussen, 1990; Zhdanov et al., 2004; Fedi, et al., 2005;Dransfield, 2010; Beiki et al., 2011). Various

platforms, such as airborne, ground, and satellites have been utilized for GGT data observation. Although, there are many boundary delineation methods that perform well in the user coordinate system for potential fields, recent research (Li, 2015) shows that rotation of the coordinate system can reveal additional information in GGT data, especially when choosing the new coordinate system via eigenvector analysis. Pedersen and Rasmussen (1990) introduced eigenvector analysis on the gradient tensor of potential field data and derived the relationship between the eigenvalues and eigenvectors of the potential

field source parameters. Beiki and Pedersen (2010) combined Euler deconvolution and the eigenvector analysis to estimate the depth of the geological body. Sertcelik and Kafadar (2012) suggested that the eigenvalues of CGGT (curvature gravity gradient tensor) could be used to detect the edges and corners of a subsurface source, while Oruc et al., (2013) applied the CGGT eigenvalue analysis on a complex field data interpretation application. Zuo and Hu (2015) proposed the eigenvalue analysis method for GGT data incorporating signal normalization. In addition to these eigenvalue methods, Cevallos (2016)

proved that using tidal tensor theory to analyze the equipotential surface can, to some extent, reach an equivalent result to the eigenvalue analysis method.

Although the utility of eigenvalues of GGT data for boundary detection performs well on source boundary detection as discussed in previous papers, the corresponding eigenvector analysis for locating the source position has not been presented. Identifying the source position has important utility not only in geologic applications, but also in environmental, time-lapse,

and engineering applications as well. In this paper, we apply eigenvector analysis on GGT data to delineate the source positions and boundaries. The relationship between the proposed eigenvector analysis method and the source parameters is developed in detail. Unlike previous Bouguer gravity anomaly boundary location methods that mainly indicate edges and CGGT methods that detect the corners, the proposed method is designed to locate the centroid of sources while delineating the boundary of them, tying edges to sources, functioning even when sources overlap. In the experimental section, both

synthetic and field data sets are used to illustrate the effectiveness of the method.

## 2 Eigenvector analysis of GGT

The gradient of the gravitational field, **T**, can be written in the form (Eq. (1)):



$$\mathbf{T} = \nabla\nabla\mathbf{U} = \begin{bmatrix} \dfrac{\partial^2 \mathbf{U}}{\partial x^2} & \dfrac{\partial^2 \mathbf{U}}{\partial x\,\partial y} & \dfrac{\partial^2 \mathbf{U}}{\partial x\,\partial z} \\[2mm] \dfrac{\partial^2 \mathbf{U}}{\partial y\,\partial x} & \dfrac{\partial^2 \mathbf{U}}{\partial y^2} & \dfrac{\partial^2 \mathbf{U}}{\partial y\,\partial z} \\[2mm] \dfrac{\partial^2 \mathbf{U}}{\partial z\,\partial x} & \dfrac{\partial^2 \mathbf{U}}{\partial z\,\partial y} & \dfrac{\partial^2 \mathbf{U}}{\partial z^2} \end{bmatrix} = \begin{bmatrix} \Phi_{xx} & \Phi_{xy} & \Phi_{xz} \\ \Phi_{yx} & \Phi_{yy} & \Phi_{yz} \\ \Phi_{zx} & \Phi_{zy} & \Phi_{zz} \end{bmatrix} \tag{1}$$

where $\Phi_{ij} = (i = x, y, z; j = x, y, z;)$ denotes the elements of the symmetric gravity gradient tensor. We assumed that the gradient tensor is decomposed as: $\mathbf{V'TV} = \Lambda$, where $\mathbf{V} = [\mathbf{v}_1\ \mathbf{v}_2\ \mathbf{v}_3]$ is the eigenvector matrix and $\Lambda = diag[\lambda_1, \lambda_2, \lambda_3]$ is the eigenvalue matrix. Eigenvalue and eigenvector decomposition has been described in many textbooks; we use the notation as

described in Anton and Rorres (2000). Pedersen and Rasmussen (1990) suggested the eigenvalue decomposition algorithm for gradient tensor data, and expressed the relationship between the eigenvalue decomposition and coordinate system rotation. Dransfield (1994) proposed a method that rotates the coordinate system around the z-axis with a rotation transform, and Cevallos (2016) introduced the expression of the eigenvector obtained from 3-D tensor data as Eq. (2) shows:

$$\mathbf{v}_i = \begin{pmatrix} \left[\Phi_{xy}\Phi_{yz} - \Phi_{xz}(\Phi_{yy} - \lambda_i)\right]\left[\Phi_{xz}\Phi_{yz} - \Phi_{xy}(\Phi_{zz} - \lambda_i)\right] \\[2mm] \left[\Phi_{xz}\Phi_{yz} - \Phi_{xy}(\Phi_{zz} - \lambda_i)\right]\left[\Phi_{xz}\Phi_{xy} - \Phi_{yz}(\Phi_{xx} - \lambda_i)\right] \\[2mm] \left[\Phi_{xy}\Phi_{yz} - \Phi_{xz}(\Phi_{yy} - \lambda_i)\right]\left[\Phi_{xz}\Phi_{xy} - \Phi_{yz}(\Phi_{xx} - \lambda_i)\right] \end{pmatrix} \tag{2}$$

Eq.(2) is theoretically derived from the classical matrix eigenvector decomposition method, and the numerical calculated results are equivalent. Pedersen and Rasmussen (1990) illustrated that the eigenvector $\mathbf{v}_1$ (corresponding to the largest eigenvalue $\lambda_1$) display a relationship between the observer and the source. They deduced the relationship based the classical gravity gradient forward equation with a 3D point mass model. Their deduced result shows that the eigenvector $\mathbf{v}_1$ indicates the direction from the observer to the point source ($\mathbf{v}_1 = (x', y', z')/R$, where $R$ is the distance between the point source and

observation point). Although the practical geological structure is much complicated than a mass point, it can be considered as a combination of many point mass cells, and the eigenvector of these sets of cells contains the boundary information of the total anomalous source. Beiki (2010) suggested that the eigenvalue decomposition could be considered as an equivalent point source method for a causative gravimetric body. Li (2015) presented theoretical and practical aspects related to the eigenvectors and rotations of the gravity gradient tensor. Li (2015) suggested that the rotation of the coordinate system

aligned with the source direction will lead to the diagonalization of a gravity gradient matrix (in the absence of noise, the gradient matrix is symmetric positive-semidefinite). The magnitude of $\Phi_{xy}$, $\Phi_{xz}$ and $\Phi_{yz}$ will be much smaller than the



diagonal terms after the transforms. He also discussed the necessity of performing such a spatially-dependent coordinate rotation to accurately calculate curvature.

For every observation point, we assume that the eigenvector $\mathbf{v}_1$ is unique, and the diagonalization can be applied for local GGT data matrix effectively. Physically, the diagonalization of the GGT data matrix can reduce the value of

components $\Phi_{xy}$, $\Phi_{xz}$ and $\Phi_{yz}$. The eigenvector decomposition on every single observation data point can extract the correct source centroid position direction, and this procedure will not be influenced by complex geological structure and noise.

In this paper, we propose the source location method according to the position vector of the equivalent source with eigenvector analysis. The position vector contains the rotated angle $\phi$ and the horizontal vector $\mathbf{r}$. As Fig. 1 shows, angle $\phi$

rotates along the z-axis with eigenvector decomposition. Physically, the essence of the eigenvector analysis is to decompose GGT data into two parts – the eigenvector dataset related with the source position, and the eigenvalue data set is related with the anomalous intensity of the source. The eigenvector in every observation position ($\mathbf{v}_1(x',y',z')'$) is a unit vector that is directly points to a source centroid, while the eigenvalue data sets contain magnitude information of the source. For example, changing a point source density can result in a different eigenvalue, but the eigenvector will not be changed. In this paper,

we utilize the eigenvector $\mathbf{v}_1$ to measure the source centroid position from every observation point in observation plane. The angle $\phi$ is used to measure the position direction along the z-axis. It is the complementary angle between the vector $\mathbf{v}_1$ that points to the source centroid and the vertical z-axis. In other words, for every observation data point, the eigenvector transform can be considered as a rotation from z-axis to z'-axis. $\mathbf{r}$ is the horizontal vector of the equivalent source to the observation point in a rotated horizontal plane, while $|\mathbf{r}|$ is the horizontal magnitude of the eigenvector in the horizontal

plane. The rotated angle $\phi$ can be identified according to the eigenvector $\mathbf{v}_1$.

As Fig.1 shows, if the buried source is immediately below the observation point, then the angle $\phi$ will reach the maximum value ($\pi/2$ radians) and $|\mathbf{r}|$ will equal zero. Thus, the rotated angle $\phi$ indicates the actual source direction from the observation point. We propose to utilize the angle $\phi$ and vector $\mathbf{r}$ to indicate the centroid and the profile of the equivalent sources.

The eigenvector $\mathbf{v}_1$ can be calculated from the gravity tensor components directly (Eq. (2)); this eigenvector should be unit vectors. However due to systematic and random errors, usually eigenvectors $\mathbf{v}_1$ calculated in this manner are not unit vectors. Therefore, we apply a balance operator $\mathbf{n}(\cdot)$ on $\mathbf{v}_1$ to reduce it to a unit vector. According to Eq. (2) and above analysis, we locate the source centroid and the outlines by using $\tan\phi$ and $\mathbf{r}$, as in Eq.(3)




$$\tan\phi = \left| \mathbf{n}\left(\left(\Phi_{xy}\Phi_{yz} - \Phi_{xz}(\Phi_{yy} - \lambda_1)\right)\left(\Phi_{xz}\Phi_{xy} - \Phi_{yz}(\Phi_{xx} - \lambda_1)\right)\right) \right| \Big/ |\mathbf{r}|$$
$$\mathbf{r} = \mathbf{n}\left(\left(\Phi_{xz}\Phi_{yz} - \Phi_{xy}(\Phi_{zz} - \lambda_1)\right)\left((\Phi_{xy}\Phi_{yz} - \Phi_{xz}(\Phi_{yy} - \lambda_1)) + (\Phi_{xz}\Phi_{xy} - \Phi_{yz}(\Phi_{xx} - \lambda_1))\right)\right) \tag{3}$$
,

Briefly, $\tan\phi$ can be also expressed as $\tan\phi = \left| \hat{\mathbf{v}}_1^z \right| \Big/ \left| (\hat{\mathbf{v}}_1^x + \hat{\mathbf{v}}_1^y) \right|$. From the view of the eigenvector, if there exists only a single

point source, $|\mathbf{r}|$ will equal 0 at the position of source centroid in the observation plane. Since $\mathbf{v}_1$ is a unit vector, $z'$ (Eq. (2))

will nearly equal 1 and $\tan\phi \to \infty$. As is known, in practical data processing and interpretation, the sources are

inhomogeneously distributed in all directions, and multiple sources commonly overlap one another, so the modulus of vector

$\mathbf{r}$ when directly over the source will not exactly equal 0. $\tan\phi$ will present a relative large value above the centroid of

sources on observation plane. With the considering of negative and positive anomaly in data, we add the vertical gravity data

$\Phi_z$ in and define *GTA* (Gradient Tensor Angle) as expressed in Eq. (4):

$$GTA = \Phi_z * \lceil \tan\phi \rceil_\beta \tag{4}$$
,

where $\Phi_z = \partial U / \partial z$. $\Phi_z$ is the spatial change rate of gravitational acceleration in the vertical direction. Comparing with

Bouguer gravity data, $\Phi_z$ can present a more significant anomalous value at the observation position above a buried source,

while the spectral power of $\Phi_z$ contains more high frequencies anomalous information. Although $\Phi_z$ commonly is not

directly measured in GGT survey, it can be calculated by using the measured vertical gravity gradient $(\Phi_{zz})$ and regional

ground gravity data (Dransfield, 2010).

Small sources, or inhomogeneous distribution of a source can present some small local $\tan\phi$ peak values. $\lceil \cdot \rceil_\beta$ is design for

filtering out these local small values. The filter threshold parameter $\beta$ can be automatically identified, as in Eq.(5)

$$\beta = \underset{\beta_i}{\arg\max}\left(\left|\partial(PeakNum(\beta_i))/\partial\beta\right|\right)$$
$$where \quad \beta_i \subset \left[\min(\tan\phi), \max(\tan\phi)\right] \tag{5}$$
,

where *PeakNum*(·) is the function to calculate the number of the local peak values in $\tan\phi$. The anomalous values of noise

and small sources have large distances with the values of primary anomalous sources in $\tan\phi$. So the parameter $\beta$ can be

identified by measuring the change tendency of the number of different local peak values in $\tan\phi$.

GTA is designed for mapping the source centroids position on the observation plane. The direct physical meaning can be

derived from the quantity $\tan\phi$. $\phi$ is the complementary angle between the vertical z-axis and eigenvector $\mathbf{v}_1$, which is the

vector from observation point to the sub-surface source centroid. A large angle $\phi$ will result in a large value of GTA. A peak

GTA value indicates that there is the position of a source centroid. A relatively large peak GTA value indicates the centroid

of a large anomalous source, while a small peak represents the centroid of a relative small local anomalous source.



According to Eq. (4), GTA value dramatically decreases with the distance increasing from the centroid position, and it will be close to zero value at the source boundary position. So with drawing the contours on the GTA map, it is possible to delineate the boundary of sources.

The value of $\tan\phi$ will decay dramatically with an increasing $|\mathbf{r}|$. In the multiple source condition, the *GTA* value of at every source centroid position will be influenced by others sources. It will not reach an infinitely large value, but it still relates with the mess of the source strongly. To proving this point, we deduce the following analysis on the basis of Eq. (4)

Assuming there is a point source $s_1$ and its horizontal direction vector is $\mathbf{r}_{s_1}$ ($\left|\mathbf{r}_{s_1}\right| \to 0$), we add another similar point source $s_2$ nearby in the same plane. This action produces an eigenvector calculation value change ($\mathbf{r}_{s_2}^d$) for $\mathbf{r}_{s_1}$. Then, the horizontal direction vector of $s_1$ can be expressed as $\mathbf{r}_{s_1}' = \mathbf{r}_{s_1} + \mathbf{r}_{s_2}^d$, where $\mathbf{r}_{s_2}^d$ ($0 < \left|\mathbf{r}_{s_2}^d\right| < 1$, $\left|\mathbf{r}_{s_2}^d\right| \ll \left|\mathbf{r}_{s_1}\right|$). Then according to Eq.(3), the $\tan\phi$ of source $s_1$ at the centroid position can be expressed as:

$$\tan\phi_{s_1}' = \frac{1 - \left|\mathbf{r}_{s_2}^d\right|}{\left|\mathbf{r}_{s_1} + \mathbf{r}_{s_2}^d\right|},\tag{6}$$

As Eq.(6) shows, with the $\mathbf{r}_{s_2}^d$ added in the calculation of Eq. (3), $\tan\phi$ will not reach infinitely large values.

Considering a condition of sources overlapping, a relatively large anomaly source $s_3$ and a small anomaly source $s_4$ with horizontal vectors $\mathbf{r}_{s_3}$, $\mathbf{r}_{s_4}$ ($\left|\mathbf{r}_{s_3}\right| \to 0$, $\left|\mathbf{r}_{s_4}\right| \to 0$), respectively. We overlap source $s_4$ on the source $s_3$. The horizontal and vertical distance between the equivalent points of $s_3$, $s_4$ are $h$ and $d$. With the sources overlapping, the fields interfere with each other. Assuming $\mathbf{r}_{s_3}^b$ and $\mathbf{r}_{s_4}^b$ ($m \cdot \left|\mathbf{r}_{s_4}^b\right| = \left|\mathbf{r}_{s_3}^b\right|$, $\left|\mathbf{r}_{s_4}^b\right| \ll \left|\mathbf{r}_{s_3}^b\right|$) are the horizontal distance changes of $\mathbf{r}_{s_3}$ and $\mathbf{r}_{s_4}$ while the two sources are overlapped. Then the ratio of $\tan\phi$ for source $s_3$, $s_4$ can be expressed as:

$$\frac{\tan\phi_{s_3}}{\tan\phi_{s_4}} = \frac{(1 - \left|\mathbf{r}_{s_3}\right| - \left|\mathbf{r}_{s_3}^d\right|)(\left|\mathbf{r}_{s_3}^d\right| + m\left|\mathbf{r}_{s_3}^d\right|)}{(\left|\mathbf{r}_{s_3}\right| + \left|\mathbf{r}_{s_3}^d\right|)(1 - \left|\mathbf{r}_{s_3}^d\right| - m\left|\mathbf{r}_{s_4}^d\right|)} \approx m,\tag{7}$$

According to Eq. (7), in the overlapping condition the $\tan\phi$ of the two sources does not interfere with each other dramatically, the $\tan\phi$ ratio of them still is equal to value $m$ approximately. Both the local maximum values of weak anomaly sources and strong anomaly sources can be easily located.



## 3  Experiments

### 3.1 Synthetic Experiments

In this section, the performance of the proposed method is tested with synthetic data. The selection of the scalar parameter ( $\beta$ ) is discussed in detail in the field example.

Considering the multiple-source scenario, we design a synthetic model that contains nine sources which are distributed in different depths, as shown in Fig.2. The depth to the top of the nine sources ranges from 1km to 2.6km, each with equal depth extents of 0.2km. The density contrast of each source is 0.3 g/cm$^3$. There are 200x200 observation points involved in this experiment, and both x-y direction lateral extent of this synthetic model are set as 100m. The data are simulated as a full-tensor gradiometer response observed at ground.

The vertical gravity data $\Phi_z$ and the tensor component $\Phi_{zz}$ are shown in Fig.3 with a 0.1km spaced observation grid. We calculate $GTA$ according to Eq. (4) with $\beta$ =10 (Fig. 4). The value of $\beta$ is identified according to Eq. (5).

As Fig.4 shows, although there are multiple sources and the anomalies are weak for the deep sources, the result of $GTA$ is not obviously influenced by this complexity. The $GTA$ value at the centroid of the 1st (shallowest) source reaches a value of 0.6098, while the $GTA$ value at the centroid of the 9th (deepest) source is 0.1443. This $GTA$ map shows all of the centroids of

these sources with high precision. The boundaries of the anomaly sources, including the boundaries of the deeper buried sources, are all delineated correctly in the contour maps.

To test the stability and robustness of the GTA map with noise, we add Gaussian noise with a standard deviation equal to 30% of the max magnitude of the tensor components to all of the gravity gradient components. The data added noise and the output $GTA$ map are shown in Fig.5.

As Fig. 5 shows, there is obvious noise in the tensor data (Fig.5), but the source location results (Fig. 6(a)) are the same as the $GTA$ map produced by the clean data (Fig. 4(a)). The noise interferences that show in the contour map (Fig. 6 (b)) mainly come from the product of the vertical gravity data ( $\Phi_z$ )(Eq. (4)), which are not amplified in the $GTA$ map calculation.

In practice, the sources are distributed in various depths and may overlap each other. Here we design another synthetic model to test the performance of the GTA method in this condition. The deeper source (label A) is buried at a depth of 2km

with depth extent 1km. Two shallow sources (label B and C) overlap on the top of source A with a depth extent of 0.2km. The three sources are joined with each other, so these synthetic models can alternatively be considered as one whole source with an undulating upper surface. The 3D view, the plan view and the $\Phi_{zz}$ data of the synthetic model are shown in Fig. 7.

The gravity gradient anomaly shown in Fig. 7(d) is simulated based on a 0.1 km regular grid, and the gradients of the synthetic model are derived according to the formulas that were proposed by Cooper (2006). We calculate the eigenvector

$\mathbf{v}_1$ , and then delineate the source centroids and outlines (Fig. 8) with the GGT data using Eq.(3) and Eq.(4).

Numerically, the values of $\tan\phi$ are related to the source size--larger sources will show larger $\tan\phi$ values. The $\tan\phi$ value



of the primary source (label A) is 6249, while the other sources are 938 and 836 (label B and label C), respectively. From an alternative perspective, the largest $\tan\phi$ value ($\tan\phi$ =6249) can be considered as the calculation result of the aggregate model (all three blocks). With decreasing threshold value $\beta$, more and more details of the buried sources are shown in the *GTA* map. It may be difficult to encapsulate the centroids and boundaries of such varied sources with one *GTA* map;

however, a series of maps (Fig. 8) with different threshold values will show the centroids and contours of varying sources clearly. This experiment also indicates that the value of $\tan\phi$ is related with the horizontal area of the source, for the only model difference between the source B and the source C is their areas in horizontal plane.

### 3.2 Field data experiment

In the field data experiment, we apply the proposed method to a high-resolution airborne gravity gradient dataset over

northeast Iowa and southeast Minnesota, U.S. (Drenth, et al., 2015). This GGT data is collected by *FTG-003* Full Tensor Gradiometer (FTG) system in 2013. The survey contains 94 east-west traverse-flying lines in 400 m apart. The field data were acquired with an 80m nominal flight height and subsequently terrain corrected (2400kg/m$^3$). The GGT data contain 481x481 observation points in the study region. The $\Phi_z$ and $\Phi_{xy}$ gravity gradient components and the Proterozoic geophysical interpretation map (Drenth et al. 2015) of the survey area are shown in Fig. 9 and Fig. 10.

In the center of the survey area, there exists an obvious low-density source (unit Ysp in Fig.10). Drenth et al. (2015) interpreted the geophysical characteristics of it as a silicic pluton. They suggested that the large-amplitude gravity response of the Decorah complex has notable geophysical similarities to Keweenawan alkaline ring complexes. In this paper, we apply the GTA method on this data set with the various $\beta$ parameter selections. The results are shown in Fig. 11.

In this experiment, the field data contain both positive and negative anomalies. As Fig. 11 shows, the *GTA* locates the

centroid of positive and negative causative sources of varying scales simultaneously, while delineating the edges of them clearly. With a smaller threshold value ($\beta$ <=80), Fig.11 (a)-(d) roughly outlines Proterozoic and part of Decorah complex sources. While using the larger threshold value ($\beta$ >160), the results indicate the centroid position and the boundaries of the primary sources in the survey area, as Fig. 11 (e)-(k) shows. The results show that there are 5671, 329, 82, 20, 4 peak values display for the parameter Beta set as 10, 40, 80,160 and 320, respectively. The number of local relative small peak value (e.g.

5671 for Beta>10) is much larger than the overall peak value (e.g. 4 for Beta>320). With the value of Beta increasing, the centroid positions of larger sources start to appear in GTA map, while the small sources, or noise are filtered out gradually. By increasing the value of $\beta$, the proposed eigenvector analysis method will become more insensitive to the presence of noise. Both of the lower density unit Ysp and the higher density gabbro of the Decorah complex can be well located in this procedure.

The parameter $\beta$ is identified according to Eq.(5) to local the primary anomalous sources. We also display the procedure of this strategy with this field data, as shown in Fig.12.





The index of the $\beta$ parameter is incremented in powers of 2. As shown in Fig.12, the maximum curvature is reached at nearly 80, which is coincident with the experimental results in Fig.11(e) that shows the clear boundary range of the two primary sources. With a $\beta$ value larger than 128, the number of detected sources is less than 10, and these points are mainly indicators of the position of the primary source centroids.

In addition, a lower bound for the $\beta$ parameter can be identified via data resolution--if the approximate depth to sources of interest is known, minimum resolvable feature width can be used to place a lower bound on the parameter. For example, in this experiment, the Proterozoic features of interest are at approximately 500 meters depth; therefore features smaller than approximately 750 meters to a kilometer are not resolvable. So the distance between the detected source centroids should not be less than 750 meters. According to this standard, the parameter $\beta$ should be set equal or larger than 80 for outputting

credible results.

### 3.2.1 Comparison with alternative boundary detection algorithms

We also compare the proposed algorithm with another GGT data boundary detection algorithm NGTE, and the usual Bouguer gravity data boundary detection algorithms on $\Phi_z$ data, such as Tilt Angle (Miller & Singh, 1994), as Fig.13 shows. As Fig. 13 (a)-(d) shows, although many of the methods commonly show good performance on the Bouguer gravity anomaly

data processing, they are not very suitable for the GGT data. Fig.13 (f) shows the $\tan\phi$ map that is defined in Eq.(3). For visibility, the maximum value is limited to 12. This map shows that there is a huge negative-density source (Ysp) in the centroid and the positive-density sources comprising the Decorah complex in the southern direction. All of the centroids and the boundary details of these bodies are well-delineated by the proposed method.

### Conclusions

We have introduced a new method to locate the centroids and boundaries of sources for full gravity gradient tensor data across different scales. Based on the eigenvector decomposition of GGT data, we describe the theoretical basis and derive the procedure of the proposed method in detail. In the context of realistic source geometries, the interference of multiple sources has been specially discussed. We use two different synthetic models to demonstrate the efficacy and robustness of the method in the presence of multiple, overlapping sources with noise. These synthetic experiments also demonstrate the

insensitivity of the method to the amplitude of the anomaly, working effectively even for weak anomalies.

We applied the technique to a field GGT dataset from northeast Lowa to test the proposed method, and two strategies to choose the scale parameter were derived and discussed in detail. The proposed method delineates source centroid positions and horizontal boundaries of sources in varying scales, for both positive and negative anomalies. Compared with conventional potential field data, our proposed method utilizes more dimensions and information that are contained in GGT

data, and provides a clear map of the buried source bodies. This method is essentially based on an eigenvector analysis, so individual small sources or random noise do not interfere significantly with the result. However, like other detection methods



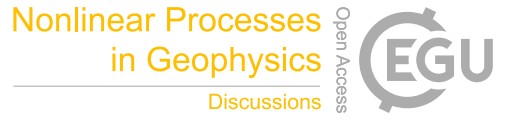

it is also difficult to clearly detect individual small sources, which is an area of future research.

Ultimately, the proposed method is feasible and reliable to locate source positions with full gravity gradient tensor data, even in areas where the sources overlap. This can be used as an effective tool for geological interpretation, locating the positions of exploration wells, or in seeding 3D density inversion algorithms (e.g. Uieda and Barbosa,2012).

## 5   Acknowledgments

This research is funded by the National Natural Science Foundation of China (No. 41630317, No. 41674110), We would like to thank Benjamin Drenth and Cericia Martinez for their valuable suggestions.

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





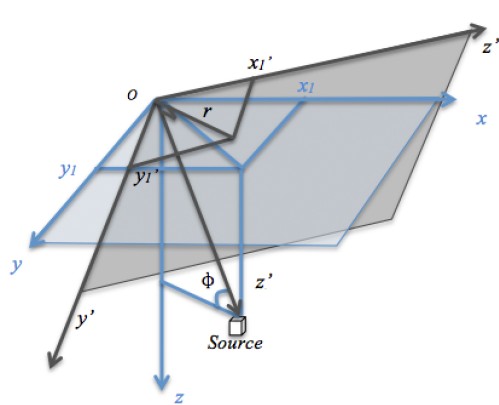

**Fig.1: The coordinate rotation and angles.**

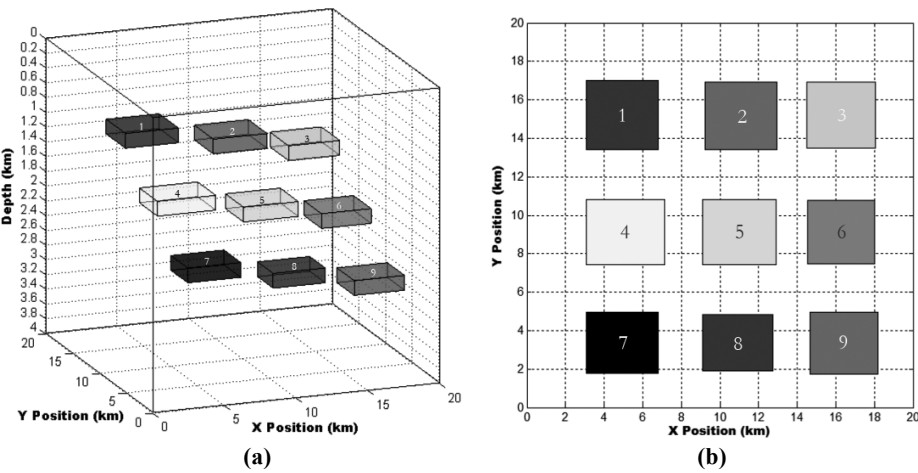

| (a) | (b) |

5  **Fig. 2: The synthetic model with nine sources (a) 3D view; (b) plan section view.**

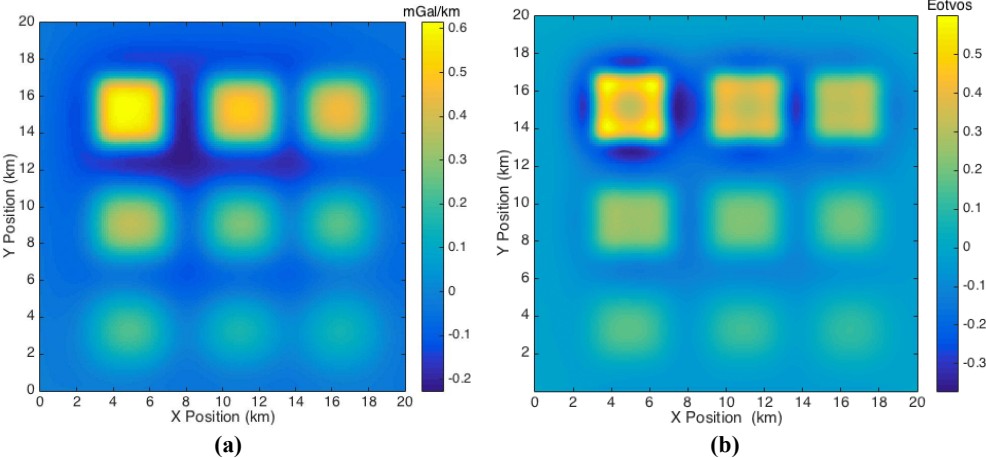

| (a) | (b) |

**Fig. 3: (a) The vertical gravity data of the model given in Fig. 2; (b) $\Phi_{zz}$ tensor component.**





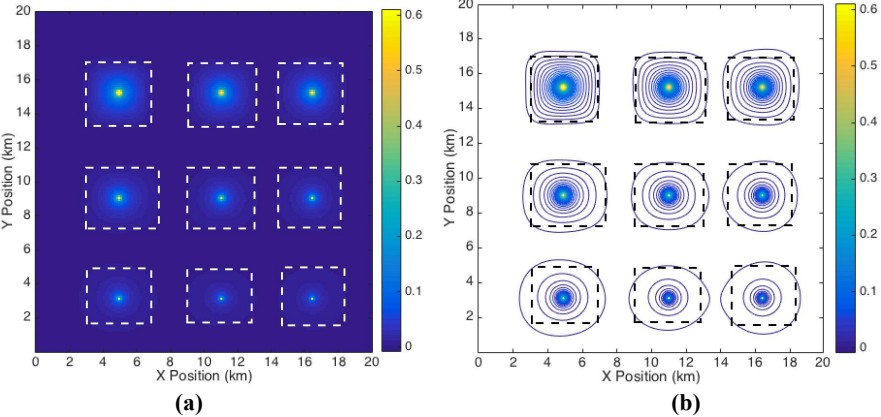

**Fig.4: (a)** *GTA* **map (Locating the centroids of sources) of simulated data in Fig.3; (b) Contour map of Figure (a). (The dash line indicates the edge of synthetic model.)**

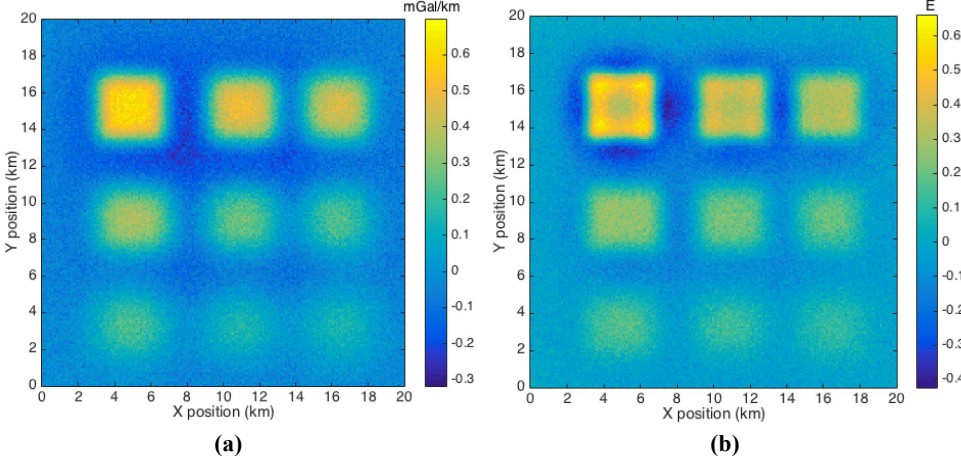

**Fig. 5: (a) The vertical gravity data ( $\Phi_z$ ) of the model given in Fig. 2 added 30% noise; (b) $\Phi_{zz}$ tensor component added 30% noise.**


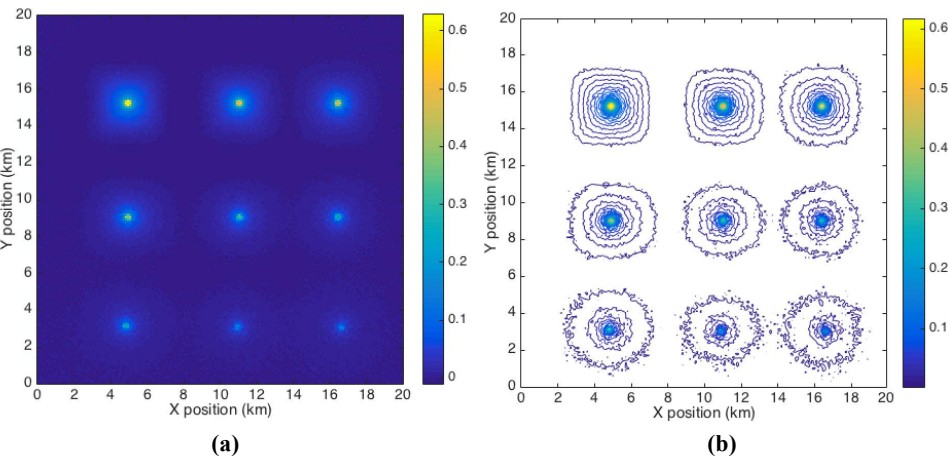

**(a)**                                   **(b)**

**Fig. 6: (a)** *GTA* **map  (Locating the centroids of sources) of simulated data in Fig.5; (b) Contour map of Fig.6 (a) for representing the boundaries of the sources.**

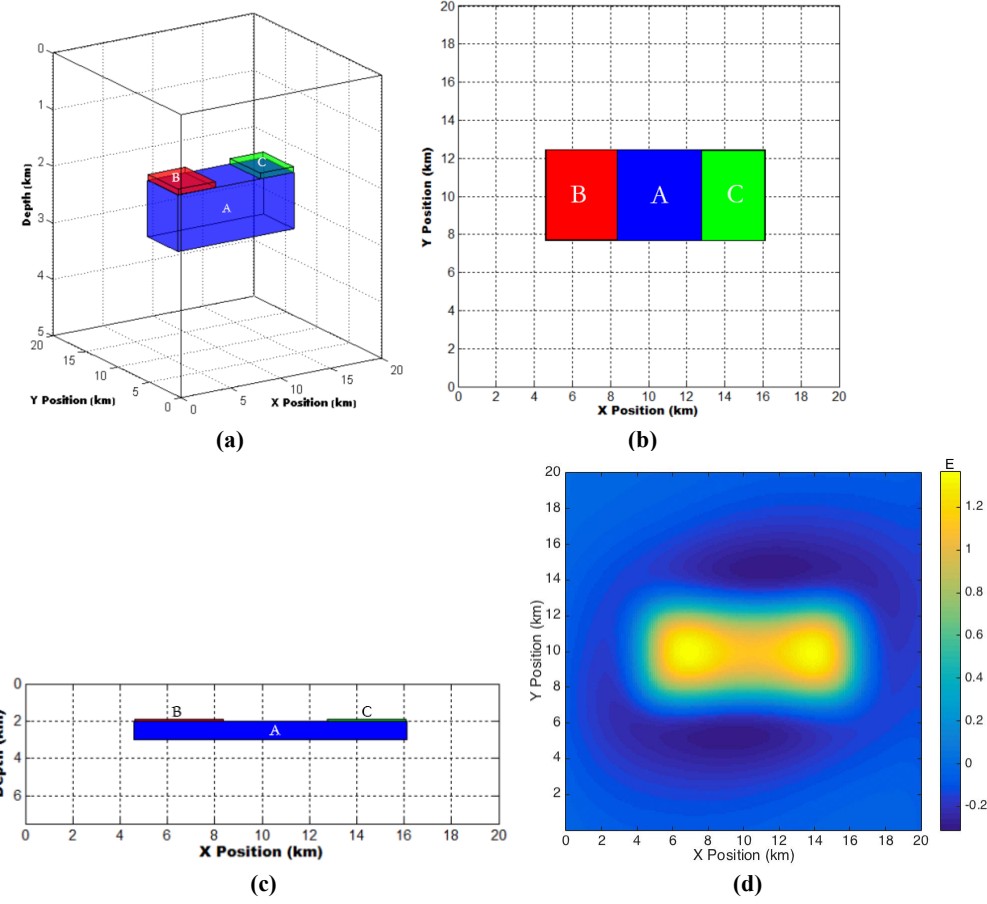

**(a)**                                   **(b)**

**(c)**                                   **(d)**

10     **Fig. 7: The synthetic model and associated vertical gravity data; (a) 3D view; (b) Plain view; (c) Cross section view; (d)** $\Phi_{zz}$ **tensor component.**





**Fig. 8: The *GTA* maps (Locating the centroids of sources) using the synthetic model in Fig. 7, and the corresponding contour maps for representing the boundaries of the sources; (a) GTA map with $\beta = 200$ ;(b) Contour map of Figure (a); (c) GTA map with $\beta$ =880;(d) Contour map of Figure (c); (e) GTA map with $\beta$ =1000 ;(f) Contour map of Figure (e).**




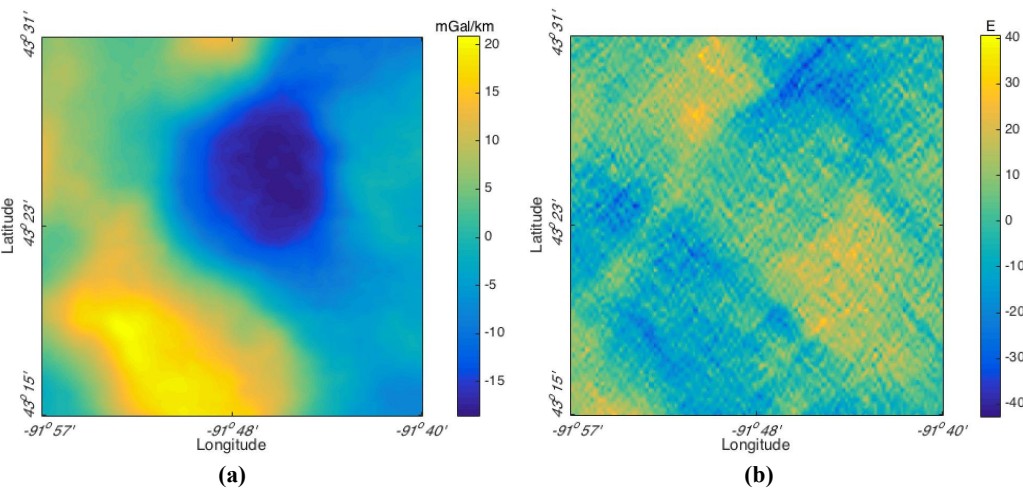

**Fig. 9 (a) The** $\Phi_z$ **data with terrain corrected using a density of 2400 kg/m3. (b)** $\Phi_{xy}$ **tensor component.(1E = 0.1 mGal/km, 1Gal=**

5 **1cm/s$^2$);**

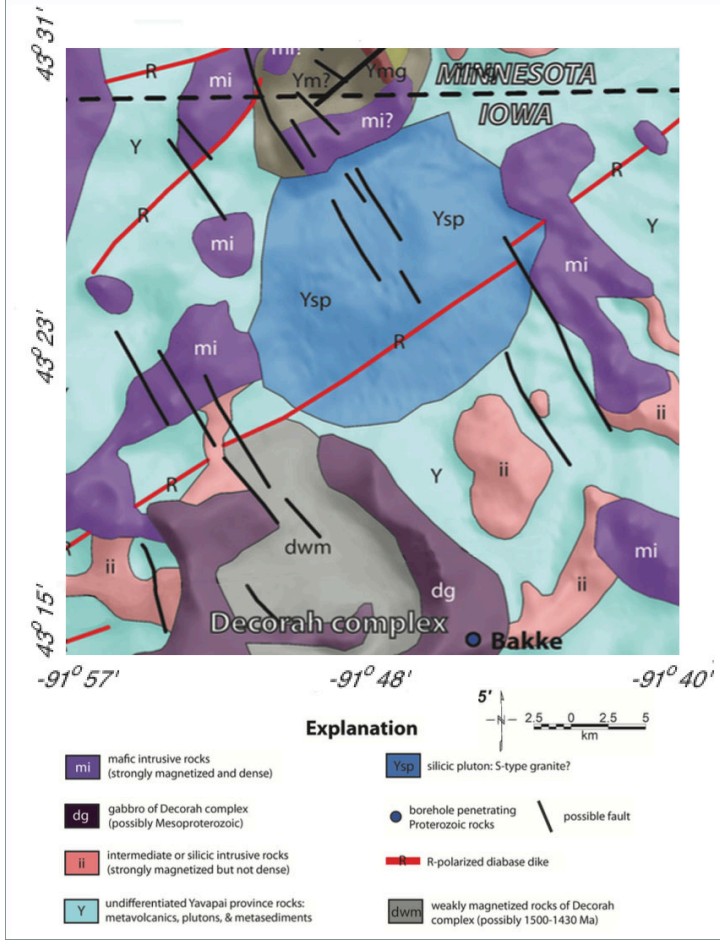





**Fig. 10 The interpretation of Proterozoic geology;   (Drenth et al., 2015);**





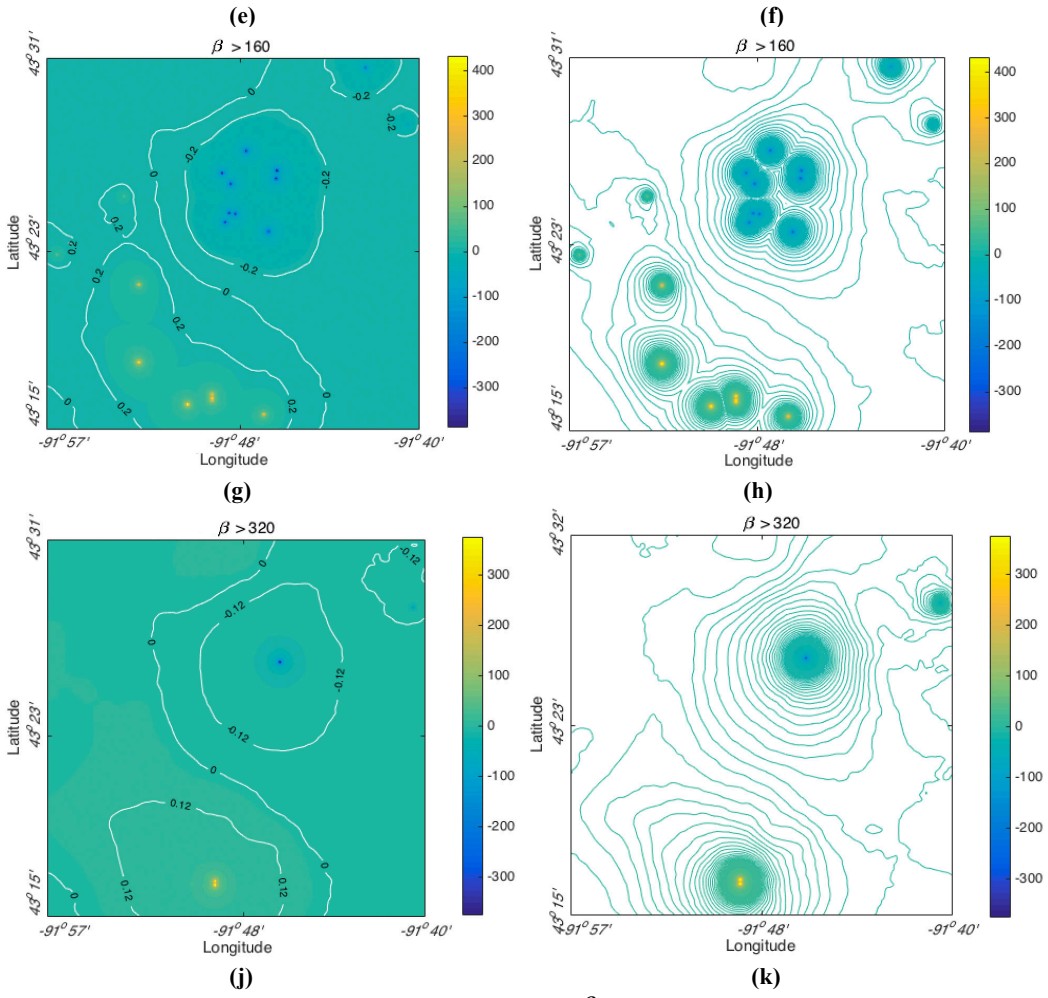

**Fig. 11 The *GTA* maps and contours map with different $\beta$ values;**





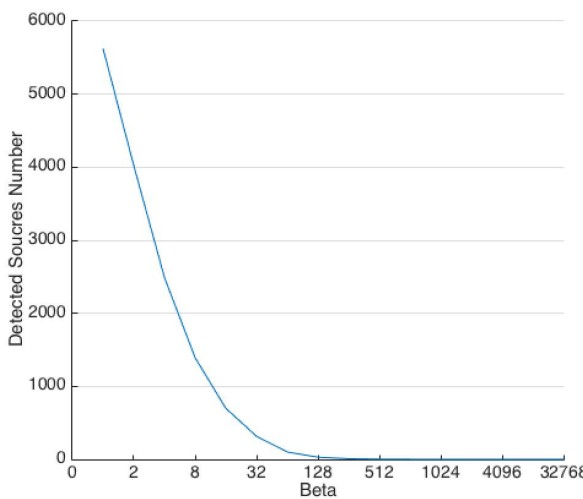

**Fig. 12  Identification of Parameter** $\beta$ ;

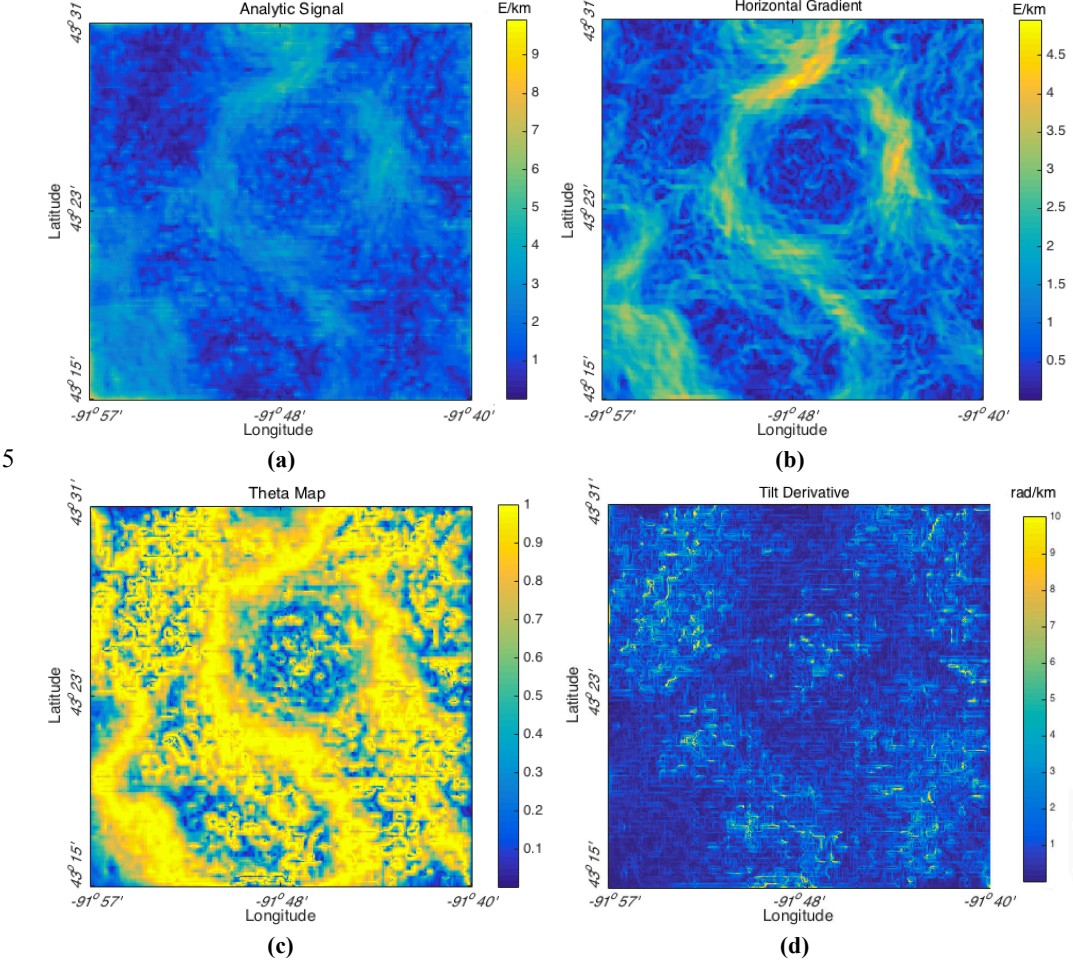

(a)

(b)

(c)

(d)





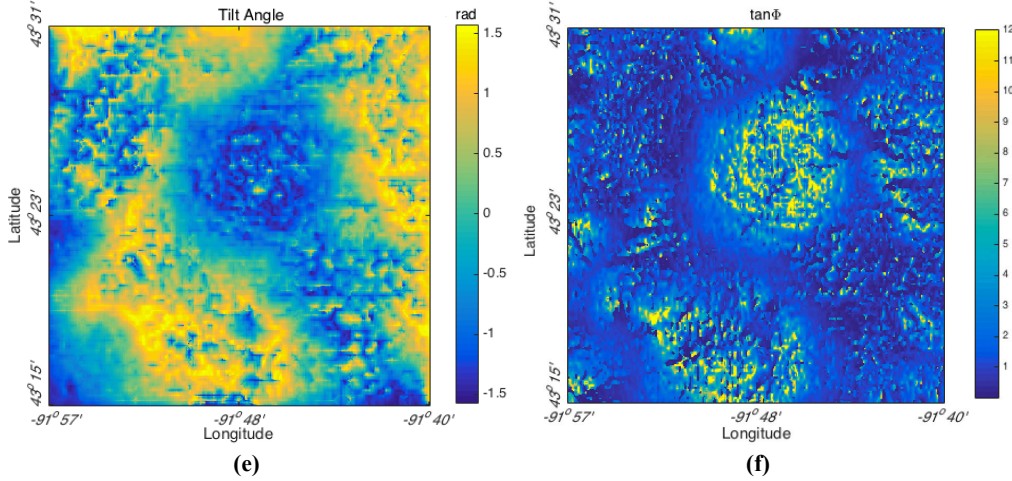

**(e)**                                                **(f)**

**Fig.13:** **(a) Analytic Signal; (b) Horizontal Gradient; (c) Theta map; (d) Tilt Derivative; (e) Tilt Angle; (f)** $\tan\phi$ **(Eq.(3));**