# Peer review of "Full-tensor gravity gradient eigenvector analysis for locating complex geological source positions"

_Nonlinear Processes in Geophysics, 2016_

## Referee Comment (RC1) · Anonymous Referee #1 · 28 Feb 2017

The manuscript gives a lengthy description of the underlying method followed by synthetic data studies and a real case using full gravity gradient tensor data.

1. I find the theoretical description much too long and too complicated. Based upon the direction of the eigenvector belonging to the largest eigenvalue of the gravity gradient tensor and the product of the vertical component and the tangent to the angle of inclination of the eigenvector a strategy is outlined to not only locate the center center of mass of a distributed source but also to outline its horizontal boundaries. Basically it is impossible to outline the boundaries using the eigenvectors. Take for example a spherical body. Using the same technique you would apparently outline the boundaries of the sphere. But this cannot be done because you need to know the density contrast of the sphere.

2. There is no reason to rotate the tensor prior to calculating the eigenvectors and eigenvalues since they are per definition independent of that. The tan(phi) can simply be expressed as the ratio between the vertical component and the horizontal component as is also indicated right below equation (3).

3. I am not convinced the location of the boundaries of isolated bodies can be made using the proposed method. Rather I think by looking at the real data example in figure 13f you have identified several "point sources" making up more complicated structures.

4. The results for synthetic data as presented in for example figure 4 are ambiguous since the contour lines have no units.

5. Compared with the results of standard methods in figure 13 I feel that the tan(phi) method is less sensitive to boundaries in agreement with the theoretical arguments given in 1.

My recommendation is to reject to manuscript due lack of significance and demonstrable errors.

---

## Referee Comment (RC2) · C. Cevallos (Referee) · 1 Mar 2017

The main problem is the definition of GTA: GTA = $\Phi Z^*[\tan\varphi]\beta \tan\varphi$ over the centres of mass of sources attains very large values.

This means that the GTA is an amplification filter of $\Phi Z$. The authors implicitly think it plots edges of sources by using its contours, but in practice there is no way to choose one contour over others. In general, $\varphi$ locates the centres of mass of the sources, and $\Phi Z$ has location and edge information. By multiplying them we lose information.

We could also define GGTA = $\Phi ZZ^*[\tan\varphi]\beta$ Following the authors scheme GGTA would then be better than the GTA in defining location and edges as $\Phi ZZ$ is much better than $\Phi Z$ in defining location and edges.

Finally, if $\beta$ = zero then GTA = $\Phi Z^*\tan\varphi$ and you obtain an amplified $\Phi Z$ from which the edge information is mostly absent and if $\beta$ = some very large value then GTA = $\Phi Z^*\beta$ and the centre of mass information is mostly absent, this means that $\beta$ is a "focussing" parameter: when far from sources it makes the GTA have information of only $\Phi Z$ as we get nearer to the top of sources it makes GTA almost totally dependent on $\varphi$. The edges get lost in this process, that is, they become dependent on $\beta$ in an unpredictable way.

My recommendation is to reject the manuscript.

---

## Author Comment (AC1) · 2 Mar 2017

Reply 1. We added an additional theoretical analysis as shown in Eq.(6) ,Eq.(7). This part is wrote to prove that the method is suitable for sources overlapping condition and $\tan\varphi$ will not be a infinitely value in practice. We think this is important for real data processing, so we make a theoretical analysis carefully. The proposed method estimate the boundary according to the value of $\tan\varphi$. It will nearly equal to 0 at the position of source boundary. The value of $\tan\varphi$ is estimate according to the data directly. So the proposed method does not need the information of the density contrast of sources.

Reply 2. Actually, we do not rotate the tensors. We use the expression of "rotation of the coordinate system" to illustrate the physical meaning of tensor eigenvector decomposition. As Beiki (2010) and Li (2015) suggestion that the eigenvector decomposition

could be considered as a "rotation of the coordinate system".

Reply 3. The practical geological structure is complex. Usually, the sharp of source is irregular, and the surface of it is unsmooth. As we know, the GGT data contains detail geological information, and more information can be extracted through algorithms. Fig.13 f does not map for illustrating the boundary of sources. It is a tanÏṬ map which is used to display the centroids and distribution of sources. Fig.11 is draw for delineate the boundary of source.

Reply 4. The contour lines in Fig.4 have color which is listed by the color bar. We defined GTA as a relative numerical measurement.

Reply 5. Fig.13 f is not listed for illustrating the boundary of sources. And for the reason of display the map clearly, we limit the maximum value of the map to 12. The contour map is displayed in Fig.11.

---

## Author Comment (AC2) · 2 Mar 2017

Reply: The purpose to introduce $\Phi Z$ in GTA is to distinguish the negative and positive anomaly in a data. As Fig.13 f. shows, $\tan\varphi$ will always be a positive value. To distinguish both of the negative and positive source in GTA, we added $\Phi z$ in. We did not design GTA as an amplification filter of $\Phi z$. Because the numerical rang of $\tan\varphi$ is relatively very large. For example, in field data experiment, $\tan\varphi$ in the rang of [0,1.472e+03]. While $\Phi z$ in the range of [-18,21] which nearly 1% of $\tan\varphi$. The main contribution of $\Phi z$ is identify the anomaly is positive of negative at a corresponding position. Yes, $\tan\varphi$ is used to local the centers of sources. But the edge information is also extracted from $\tan\varphi$. $\tan\varphi$ display a peak value at the source center, and it will also display as a relative small value which in the position nearby the source centers. So we utilize these small $\tan\varphi$ values to delineate the contours of sources. Yes, $\Phi zz$

provides more detail information of source than $\Phi z$. But in this research, for the goal of distinguishing the negative and positive source, $\Phi z$ can provide enough information. Thanks for your valuable suggestion, in further research, we want add $\Phi zz$ in and utilizes it to extract more detail source information.

---

## Short Comment (SC1) · 14 Mar 2017

Have worked on some gravity gradient data before. From the practical point of view, my feeling is sometimes the noise filters used in the processing were kind of subjective. If there is too much noise, some methods may fail. Looking at the figures, it seems the new approach is not subject to this problem? If this can be confirmed or clarified, it may be a useful tool. Some interesting questions in the comments above. Not sure how important the phi_z is in the final result, but the field data set seems working well? Further explanation would be nice.

---

## Author Comment (AC3) · 14 Mar 2017

After a serious consideration, I think your suggestions are right. It is impossible and inappropriate to outline the source boundaries using the eigenvectors. As you mentioned, there are lots of problems to delineate source boundary without know the density contrast. I have deleted the whole contents of boundary detection and revised the related contents. I have revised the title of this manuscript. I have added units on Fig.4 and related Figures (Fig.4, Fig.6, Fig.8, Fig.11). And I have submit a new revised manuscript on NPG website.

Please also note the supplement to this comment:
http://www.nonlin-processes-geophys-discuss.net/npg-2016-75/npg-2016-75-AC3-supplement.pdf

[Figure]

[Figure]

**Supplement:**

[revised manuscript text omitted]

---

## Author Comment (AC4) · 14 Mar 2017

Your comment is correct that the boundary detection section in the previous manuscript has problems. And I did not illustrate dz clearly in the previous manuscript: I wrote "Comparing with Bouguer gravity data, dz can present a more significant anomalous value at the observation position above a buried source, while the spectral power of dz contains more high frequencies anomalous information." This is not right. I have deleted this. I have deleted the section of boundary detection, revised the related contents and the title of this manuscript. I submit a new revised manuscript to you.

Please also note the supplement to this comment:
http://www.nonlin-processes-geophys-discuss.net/npg-2016-75/npg-2016-75-AC4-supplement.pdf

---

## Author Comment (AC5) · 15 Mar 2017

[revised manuscript text omitted]
 |\mathbf{r}_{s_4}^b| = |\mathbf{r}_{s_3}^b|$, $|\mathbf{r}_{s_4}^b| \ll |\mathbf{r}_{s_3}^b|$) are the horizontal distance changes of $\mathbf{r}_{s_3}$ and $\mathbf{r}_{s_4}$ while the two sources are overlapped. Then the ratio of $\tan\phi$ for source $s_3$, $s_4$ can be expressed as:

15    $$\frac{\tan\phi_{s_3}}{\tan\phi_{s_4}} = \frac{(1-|\mathbf{r}_{s_3}|-|\mathbf{r}_{s_3}^d|)(|\mathbf{r}_{s_3}^d|+m|\mathbf{r}_{s_3}^d|)}{(|\mathbf{r}_{s_3}|+|\mathbf{r}_{s_3}^d|)(1-|\mathbf{r}_{s_3}^d|-m|\mathbf{r}_{s_4}^d|)} \approx m \tag{7}$$

,

According to Eq. (7), in the overlapping condition the $\tan\phi$ of the two sources does not interfere with each other dramatically, the $\tan\phi$ ratio of them still is equal to value *m* approximately. Both the local maximum values of weak anomaly sources and strong anomaly sources can be easily located.

**3 Experiments**

20   ### 3.1 Synthetic Experiments
In this section, the performance of the proposed method is tested with synthetic data. The selection of the scalar parameter ($\beta$) is discussed in detail in the field example.

Considering the multiple-source scenario, we design a synthetic model that contains nine sources which are distributed in

different depths, as shown in Fig.2. The depth to the top of the nine sources ranges from 1km to 2.6km, each with equal depth extents of 0.2km. The density contrast of each source is 0.3 g/cm$^3$. There are 200x200 observation points involved in this experiment, and both x-y direction lateral extent of this synthetic model are set as 100m. The data are simulated as a full-tensor gradiometer response observed at ground.

5  The vertical gravity data $\Phi_z$ and the tensor component $\Phi_{zz}$ are shown in Fig.3 with a 0.1km spaced observation grid. We calculate *GTA* according to Eq. (4) with $\beta$ =10 (Fig. 4). The value of $\beta$ is identified according to Eq. (5).

As Fig.4 shows, although there are multiple sources and the anomalies are weak for the deep sources, the result of *GTA* is not obviously influenced by this complexity. The *GTA* value at the centroid of the 1$^{st}$ (shallowest) source reaches a value of 73.0557, while the *GTA* value at the centroid of the 9$^{th}$ (deepest) source is 8.673. This *GTA* map shows all of the centroids of

10  these sources with high precision.

To test the stability and robustness of the GTA map with noise, we add Gaussian noise with a standard deviation equal to 30% of the max magnitude of the tensor components to all of the gravity gradient components. The data added noise and the output *GTA* map are shown in Fig.5.

As Fig. 5 shows, there is obvious noise in the tensor data (Fig.5), but the source centroid location results (Fig. 6(a)) are the

15  same as the *GTA* map produced by the clean data (Fig. 4(a)). The noise interferences that show in the contour map (Fig. 6 (b)) mainly come from the product of the vertical gravity data ( $\Phi_z$ )(Eq. (4)), which are not amplified in the *GTA* map calculation.

In practice, the sources are distributed in various depths and may overlap each other. Here we design another synthetic model to test the performance of the GTA method in this condition. The deeper source (label A) is buried at a depth of 2km with depth extent 1km. Two shallow sources (label B and C) overlap on the top of source A with a depth extent of 0.2km.

20  The three sources are joined with each other, so these synthetic models can alternatively be considered as one whole source with an undulating upper surface. The 3D view, the plan view and the $\Phi_{zz}$ data of the synthetic model are shown in Fig. 7.

The gravity gradient anomaly shown in Fig. 7(d) is simulated based on a 0.1 km regular grid, and the gradients of the synthetic model are derived according to the formulas that were proposed by Cooper (2006). We calculate the eigenvector $\mathbf{v}_1$, and then locate the source centroids (Fig. 8) with the GGT data using Eq.(3) and Eq.(4).

25  Numerically, the values of $\tan\phi$ are related to the source size--larger sources will show larger $\tan\phi$ values. The $\tan\phi$ value of the primary source (label A) is 8290, while the other sources are 1245 and 1109 (label B and label C), respectively. As we analyze in Eq.(6), a single source may produce a large value $\tan\phi$, if it is less influenced by other sources. From an alternative perspective, the largest $\tan\phi$ value ( $\tan\phi$ =8290) can be considered as the calculation result of the aggregate model (all three blocks). With decreasing threshold value $\beta$, more and more details of the buried sources are shown in the

30  *GTA* map. It may be difficult to encapsulate the centroids of such varied sources with one *GTA* map; however, a series of maps (Fig. 8) with different threshold values will show the centroids of varying sources.

**3.2 Field data experiment**

In the field data experiment, we apply the proposed method to a high-resolution airborne gravity gradient dataset over northeast Iowa and southeast Minnesota, U.S. (Drenth, et al., 2015). This GGT data is collected by *FTG-003* Full Tensor Gradiometer (FTG) system in 2013. The survey contains 94 east-west traverse-flying lines in 400 m apart. The field data were acquired with an 80m nominal flight height and subsequently terrain corrected (2400kg/m$^3$). The GGT data contain 481x481 observation points in the study region. The $\Phi_z$ and $\Phi_{xy}$ gravity gradient components and the Proterozoic geophysical interpretation map (Drenth et al. 2015) of the survey area are shown in Fig. 9 and Fig. 10.

In the center of the survey area, there exists an obvious low-density source (unit Ysp in Fig.10). Drenth et al. (2015) interpreted the geophysical characteristics of it as a silicic pluton. They suggested that the large-amplitude gravity response of the Decorah complex has notable geophysical similarities to Keweenawan alkaline ring complexes. In this paper, we apply the GTA method on this data set with the various $\beta$ parameter selections. The results are shown in Fig. 11.

In this experiment, the value of $\tan\phi$ map distributes in the range of [0,1.472e+03]. While the vertical gravity data ($\Phi_z$) contains both positive and negative anomalies within the numerical range of [-18, 21](mGal/km). As Fig. 11 shows, the *GTA* locates the centroids 
[revised manuscript text omitted]